DATA RELEASE

# Tick abundance, diversity and pathogen data collected by the National Ecological Observatory Network

Sara H. Paull[1,*], Katherine M. Thibault[1] and Abigail L. Benson[2]

1 Battelle, National Ecological Observatory Network, 1685 38th Street, Boulder, CO 80301, USA
2 U.S. Geological Survey, Core Science Systems, Denver Federal Center, Building 810, Denver, CO 80225, USA

## ABSTRACT

Cases of tick-borne diseases have been steadily increasing in the USA, owing in part to tick range expansion, land cover and associated host population changes, and habitat fragmentation. However, the relative importance of these and other potential drivers remain poorly understood within this complex disease system. Ticks are ectotherms with multi-host lifecycles, which makes them sensitive to changes in the physical environment and the ecological community. Here, we describe data collected by the National Ecological Observatory Network on tick abundance, diversity and pathogen infection. Ticks are collected using drag or flag methods multiple times in a growing season at 46 terrestrial sites across the USA. Ticks are identified and enumerated by a professional taxonomist, and a subset of nymphs are PCR-tested for various tick-borne pathogens. These data will enable multiscale analyses to better understand how drivers of tick dynamics and pathogen prevalence may shift with climate or land-use change.

**Subjects** Ecology, Biodiversity, Taxonomy

**Submitted:** 28 February 2022

\* Corresponding author. E-mail: paull@battelleecology.org

Preprint submitted at https://doi.org/10.5524/prepub-DRR-202202-08

Included in the series: *Vectors of human disease* (https://doi.org/10.46471/GIGABYTE_SERIES_0002)

## DATA DESCRIPTION

### Context

This dataset contains information on tick abundance, diversity and pathogen infection collected between 2014 and 2020 by the National Ecological Observatory Network (NEON) [1, 2] at terrestrial sites across the USA, excluding Hawaii (Figure 1). Data are collected as a part of the NEON program, whose mission is to provide continental-scale, long-term, open-access ecological data for understanding ecosystem change [3]. NEON includes sampling of ticks as part of its terrestrial observations program because of their medical relevance as well as their important ecological roles [4].

Ticks pose a significant public health threat in the USA, and are responsible for transmitting many bacterial, viral and protozoan pathogens to humans and domestic animals. Tick-borne pathogens account for 95% of the ~50,000 cases of locally acquired vector-borne disease cases reported to the US Centers for Disease Control (CDC) [5]. Between 2004 and 2016, tick-borne infections more than doubled, from 22,527 to 48,610 [6]. These patterns are partly associated with broad-scale drivers such as land use and climate change; however, the specific mechanisms and geographic variability in drivers remain poorly understood [7]. There is an urgent need for more surveillance and long-term

**Figure 1.** Map of the location of the 47 terrestrial sites where NEON sampling occurs. Sites with a large green circle are core sites while the smaller gray circles represent gradient sites. Gray lines delineate the 20 eco-climatic divisions across the NEON sampling area [3]. Tick sampling is not conducted at the site in Hawaii.

datasets that can help fill gaps in distribution maps and inform models for predicting epidemiological trends [8].

As well as their importance for public health, ticks are effective sentinels of ecosystem changes, and they also affect ecological interactions. Ticks encounter many host species, creating complex interaction networks [9] that highlight their ecological connectivity and sensitivity to perturbations [4]. As ectotherms that spend most of their lifecycle detached from their hosts, ticks are highly sensitive to changes in abiotic factors such as humidity and temperature [10]. Ticks also respond to land-use change via its effects on the availability of suitable microhabitats and host community composition [11]. In addition to their dynamic responsiveness to abiotic and biotic changes, ticks themselves may help to shape ecological interactions. For instance, parasites are known to alter the behavior or mortality of their hosts with cascading effects on community dynamics [12]. There is a growing body of literature on the ecological consequences of parasitism [13, 14], and the role of ticks remains open for exploration.

This unique dataset represents a key resource for public health and ecology researchers by providing multiscale, long-term data collected using a standardized sampling framework [15], alongside co-located measurements of various related abiotic and biotic variables. This dataset meets the gold standard of tick surveillance by providing information on the distribution and abundance of tick vectors, as well as the prevalence of tick-borne pathogens [8]. Unfortunately, few existing datasets achieve this standard and gaps in data availability have hampered attempts to build a synthetic understanding of the complex interactions driving variations in tick-borne disease risk. A recent survey of state and local public health and vector control agencies found that fewer than half used active tick surveillance methods, and most of those were only measuring tick presence [16]. Furthermore, several historical tick records have been lost because of incomplete spatiotemporal, quantitative or life stage information, and whole datasets have been lost

owing to obsolete storage methods [17]. These issues highlight the importance of publishing open access standardized tick datasets.

Various related datasets can be paired with the NEON tick and tick pathogen data to either expand the spatiotemporal scope of the records or provide a rich ecosystem-scale context to the observations. For instance, the ArboNET arbovirus surveillance system, which was created by the US Centers for Disease Control (CDC) in 2000, has expanded to include tick surveillance data beginning in 2019 [18] and can be made available to researchers upon request. Several other vector databases exist; however, most primarily contain data on mosquitoes and their pathogens (e.g., [19]). The data described here were collected using drag and flag methods, which are the most widely used methods for tick surveillance projects [8], facilitating cross-comparison with other datasets [4]. NEON also offers 180 open-access datasets describing ecosystem-level variables at sites co-located with the tick surveillance plots that are available from the NEON data portal [20]. These offer unprecedented opportunities to explore cross-scale effects of abiotic and biotic factors on tick abundance, diversity, and infection patterns across environmental gradients. Some NEON datasets that are likely to be of particular interest for tick researchers include (i) observational field sampling of small mammals and their associated tick-borne pathogens, breeding land birds, and vegetation structure; (ii) instrumented measurements of variables such as temperature (soil, air, or infrared/surface), precipitation and relative humidity; and (iii) airborne remote-sensing observations of variables including normalized difference vegetation index (NDVI) and laser imaging, detection and ranging (LiDAR). Note that rodent pathogen data focused on hantaviruses through 2019, and switched to monitoring tick-borne pathogens in 2020 [6].

Several outstanding questions remain in the field of tick-borne disease ecology, and this dataset is well poised to provide insight into these knowledge gaps. For instance, poleward expansion of ticks and increases in tick abundance are predicted to result from climate change [21, 22]; however, few studies of long-term datasets have detected a clear climate signature [10]. NEON tick collection methods were explicitly designed to detect such a climate signature as the time series grows in length [4]. Ticks occur across a broad habitat range, with variable composition of host communities such that the drivers of population and infection dynamics change across space [7]. These complex and variable relationships require the use of datasets that enable analyses of patterns across multiple scales [7, 11], such as the continental-scale data provided by NEON. This dataset also facilitates research into the circumstances surrounding the introduction and spread of invasive ticks, such as *Haemaphysalis longicornis* [23], which first appears in this dataset in 2017. Public health responses to the tick-borne disease threat would also be enhanced by improvements to tick distribution maps and a better understanding of the phenology of host-seeking behavior in ticks [8], both of which are enabled by this dataset. Finally, this dataset references several archived tick specimens that could be used to quickly determine the historical distribution and abundance in the event of the discovery of a novel tick-borne pathogen [8].

## METHODS

### Tick field collections

NEON sites span continental USA, Alaska, Hawaii, and Puerto Rico. A statistical algorithm using a range of ecoclimatic factors, such as elevation, temperature, precipitation and soil characteristics [24], partitioned this area into 20 ecoclimatic domains with distinct climate,



vegetation, and ecosystem dynamics [25]. Each domain has a single core site in a wildland area, and most have one to several gradient sites intended to capture ecological or land-use gradients. There are six tick sampling plots within each of the 46 terrestrial NEON sites where tick sampling occurs (excluding Hawaii). Plots are distributed across National Landcover Database (NLCD) classes in proportion to the area within the site covered by that NLCD class. Tick plots are separated by at least 500 m and are not bisected by streams. Sampling occurs along the 160-m perimeter of each 40 m × 40 m square tick plot at all sites except at the Guanica Forest (GUAN) site in Puerto Rico. At GUAN, dense vegetation precludes the use of the square plot design, so transect paths covering 80–160 m of discontinuous segments are used instead.

Tick collection occurs during the growing season at a site. Sampling begins within 2 weeks of green-up and ends within 2 weeks of senescence. This means that ticks are typically not sampled between November and February, and the longest collection windows range from March to mid-October. The frequency of tick collection varies with the quantity of ticks collected at a site. Sites are considered high intensity when more than five ticks are collected within a 365-day window, and they are sampled once every 3 weeks. Low-intensity sites (where fewer than five ticks per year are collected) are sampled once every 6 weeks. Sampling is only performed if the ground is dry and the high temperature on the two consecutive days before sampling is above 0 °C. Sampling is also avoided during the hottest part of the day, and when winds are high enough to prevent the cloth from staying flat to the ground.

Tick sampling involves collection of ticks using the drag or flag sampling methods, which are commonly used for tick collection and recommended by the CDC for tick surveillance [26, 27]. To complete the drag method, a 1-m × 1-m cloth is pulled slowly (~one step per second) along the ground, taking care to keep the cloth flat and completely in contact with the ground. Dragging stops for collection of ticks from the cloth every 5–10 m. Dragging is the preferred sampling method, but in the case of dense vegetation where dragging does not allow the cloth to touch the ground, flag sampling is conducted instead. Flag sampling involves waving the cloth over and around vegetation, as well as sweeping underneath vegetation whenever possible. Flagging stops every 3–4 sweeps to collect ticks from the cloth. A plot must be sampled for at least 80 m to constitute a viable sample. If weather or obstacles (e.g., flooding) prevent sampling to this 80-m threshold, the data are not retained and re-sampling is attempted later, if logistically feasible. Safety recommendations for field technicians include wearing light-colored clothing, tucking pants into socks, and performing regular tick checks. If insect repellent is used, it is applied away from equipment and at least 30 minutes before field sampling, and hands are washed before handling sampling equipment.

Ticks are collected from the cloth using forceps and placed directly into vials filled with 95% ethanol. When large numbers of larvae are present, weak painter's tape may be used for more efficient collection. The tape is then soaked in ethanol overnight before removing the larvae into a vial. Any ticks found on the field technician conducting the sampling during the course of the bout are included in the vial of tick samples. Vials with samples are placed into a cooler with ice packs for transit back to the laboratory [15].

## Tick taxonomic identifications

Ticks are sent to an external laboratory for identification by a professional taxonomist. The number of ticks within each life stage is recorded separately for each bout (plot) of tick



sampling, with the sex of adult ticks also being recorded. All non-larvae are then identified to the lowest taxonomic level possible (typically species). When larval numbers exceed 200, a volumetric sampling method is used to estimate the total number of larvae in the sample. This method involves marking 1-mL vials with a line representing a volume of 200 larval ticks. Larvae submerged in ethanol are then poured into these calibrated vials and allowed to settle. The line at which the larvae settle is then marked on the vial and measured. The proportional measurements (in mm) of the two markings are multiplied by 200 to obtain the total number of ticks, such that the equation used to calculate the number of ticks is: $200 \times X/Y$ , where $X$ is the height (in mm) of the ticks in the vial, and $Y$ is the height (in mm) of where 200 ticks would fall on the vial. All collected tick samples are archived, with the exception of any nymphs that will be tested for pathogens. These archived tick samples can be requested for various uses, including further identification of larvae or genetics studies [28].

Some deviations from this protocol occurred in the earlier years of tick data collection. Specifically, from 2013–2018, there was a limited amount of unreported subsampling to meet invoice limits for identification that resulted in the potential for incomplete enumeration of ticks from a small subset of samples. Additionally, between 2016 and 2018, larvae counts were capped at 500 individuals, so even samples with >500 larvae are reported as having a total of 500.

## Tick pathogen testing

When suitable numbers of nymphal ticks from the genus *Ixodes* or *Amblyomma* are available beyond that needed for archiving, they are sent to a pathogen analysis laboratory where they are tested for infection by various pathogens. *Ixodes scapularis* nymphs are targeted for testing of *Anaplasma phagocytophilum, Babesia microti, Ehrlichia muris*-like agent, *Borrelia burgdorferi* sensu lato, *Borrelia miyamotoi,* and *Borrelia mayonii*. Nymphs in the genus *Amblyomma* are targeted for testing of *Francisella tularensis, Rickettsia rickettsii, Anaplasma phagocytophilum, Ehrlichia chaffeensis, Ehrlichia ewingii,* and *Borrelia lonestari*. Nymphal ticks are targeted for testing because most human infections with tick-borne pathogens are thought to result from bites by infected nymphs [29]. As a result, the abundance of infected nymphs is often used as a surrogate measure for tick-borne disease risk [29].

Nymphs are individually tested for each pathogen using triplex Taqman real-time PCR (polymerase chain reaction) analyses with a required method detection limit of 5–0.005 pg pathogen per 15 μL extracted sample [28]. Each tick is homogenized with 301 μL of a solution of Proteinase K (1 μL of 50 μg/μL) and tissue and cell lysis solution (300 μL) for DNA extraction. *Ixodes* and non-*Ixodes* ticks are all tested with three triplex assays. The exact assays used depend on tick genus (Table 1). Reaction mixtures are prepared using Agilent Brilliant III Ultra-Fast QPCR (Agilent, Santa Clara, CA), while the primers and dual-labeled probes are ordered from IDTDNA (Integrated DNA Technologies, Coralville, IA). Amplification is performed on an Agilent Mx3000P qPCR machine under the following conditions: one cycle at 95 °C for 10 minutes, followed by 40 cycles of 95 °C for 15 seconds and 60 °C for 60 seconds. Each plate contains a positive control and two negative controls (one extraction control and one master mix control).

Table showing the pathogens detected by each multiplex tick pathogen detection assay, which assay is used for ticks in different genera. The genes targeted by each assay as well as the label dye used are also included.

**Table 1.** Tick pathogen detection assays

| Tick type | Detection | Multiplexing | Target gene | Probe label dye |
|---|---|---|---|---|
| Hard ticks | DNA QC internal control<br>*Borrelia spp.*<br>*Ixodes pacificus* | Hard tick<br>Triplex 1 | 16S mtDNA<br>16S<br>CO1 | FAM<br>HEX<br>Cy5 |
| *Ixodes* ticks | *Borrelia burgdorferi*<br>*Borrelia miyamotoi*<br>*Borrelia mayonii* | *Ixodes*<br>Triplex 1 | ospA<br>glpQ<br>ospC | FAM<br>HEX<br>Cy5 |
| *Ixodes* ticks | *Babesia microti*<br>*Anaplasma phagocytophilum*<br>*Ehrlichia muris*-like agent | *Ixodes*<br>Triplex 2 | Tubulin<br>MSP-2<br>P13 | FAM<br>HEX<br>Cy5 |
| Non-*Ixodes* ticks | *Ehrlichia chaffeensis*<br>*Borrelia lonestari*<br>*Ehrlichia ewingii* | Non-*Ixodes*<br>Triplex 1 | dsb<br>glpQ<br>dsb | FAM<br>HEX<br>Cy5 |
| Non-*Ixodes* ticks | *Francisella tularensis*<br>*Rickettsia rickettsii*<br>*Anaplasma phagocytophilum* | Non-*Ixodes*<br>Triplex 2 | ISFtu2<br>RRi6<br>MSP-2 | FAM<br>HEX<br>Cy5 |

## DATA VALIDATION AND QUALITY CONTROL

Several data validation and quality control (QC) procedures are conducted at each step of data collection, from the field to the pathogen testing facility [30]. Quality assurance (QA) begins with a QA/QC checklist for completion in the field and domain facility immediately following sampling. Field checks include confirming accurate metadata (e.g., plot ID) and that the sampled area falls within the limit of 80–180 $m^2$. Laboratory checks include confirming that six plots are sampled at each bout within a site, and that archive vial labels are accurate. Additionally, there are quality measures at the point of data entry via the use of constrained values, pre-populated location data, and dropdown menus in the mobile application used for entering the tick collection data.

QA checks are also performed by the identification laboratory to verify accurate enumeration and identification of subsamples. A minimum of 3% of samples are counted twice by the identification laboratory and the percent difference in enumeration (PDE) is reported as:

$$|(\text{count}_1 - \text{count}_2)|/(\text{count}_1 + \text{count}_2) \times 100.$$

When the PDE is >0.05%, the results of the second count are reported. The percent taxonomic disagreement (PTD) is also for subsamples that are identified twice by the laboratory. The PTD is calculated as:

$$(1 - \text{agreements}/N) \times 100$$

where 'agreements' is the number of consistent identifications (species and sex) in the subsample and $N$ is the total number of individuals.

There is an additional set of controls at the point of ingest of data into the NEON database (Figure 2). For field-collected data, each data entry field has a set of entry validation rules, which are checked by the parser before the data will be ingested into the system.

Several QA checks are also used to verify the accuracy of pathogen testing results. A minimum of 3% of samples are tested to verify pathogen detection using a known positive control. A minimum of 3% of samples are also checked to verify that PCR is correctly identifying the absence of pathogens using a negative control for master mix. Batches of



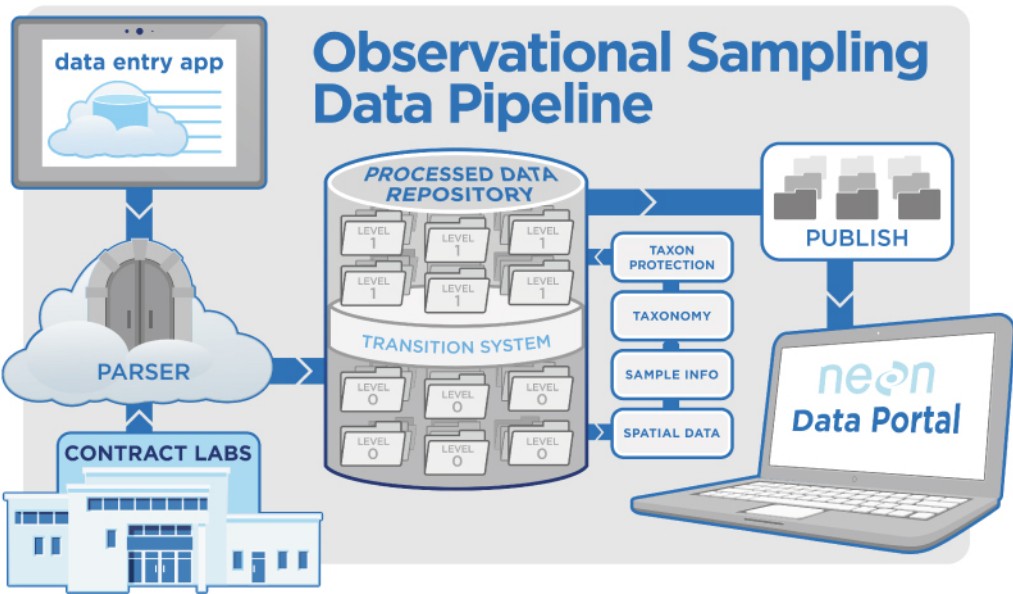

**Figure 2.** Graphic displaying the flow of NEON data from the point of collection to the point of publication. Data are either entered into data entry applications by NEON field technicians or uploaded directly by contracting laboratories, such as the facilities in which taxonomic identification and pathogen testing occur. When these data are returned to NEON Headquarters, they are interpreted by the parser. The parser performs automated quality control checks for data validation, such as ensuring that a sample returned with pathogen testing data already has a matching sample with the identification data in the database. Each NEON protocol has its own set of machine-readable instructions for data validations. NEON scientists are notified when data are rejected, allowing review of the issue. After passing through the parser, data enter the processed data repository from where they are published to the portal and freely accessible to end users.

samples are also checked for contamination via a negative control on nucleic acid extraction.

## REUSE POTENTIAL

This unique dataset offers potential for reuse by epidemiologists, ecologists, medical entomologists, biologists and others. These quality-controlled, continental-scale data are collected according to a standardized protocol and provide a complete set of information on tick abundance, diversity and pathogen infection that is challenging to find in other open-access datasets. The existence of a rich set of related ecological and environmental data collected from the same sites over a similar timespan enables novel possibilities for exploring geographic variability in the drivers of tick dynamics and infection. There is ample opportunity to fill gaps in our understanding of tick demography, distribution, host associations, and weather/climate relationships. The catalog of archived samples adds further options to expand the utility of the dataset by enabling research into such topics as taxonomy, population genetics or historical distribution of novel pathogens [31].

## DATA AVAILABILITY

The tick dataset described here is freely available through the Global Biodiversity Information Facility (GBIF), with terms cross-walked to Darwin Core [32]. The GBIF data are accessible at the GBIF National Ecological Observatory Network publisher portal [33] under a CC0 public domain waiver. These data are available in a different format from NEON,

including a dataset on tick collections and identifications [1] and tick pathogens [2]. The NEON data, through the end of 2020, include the same records as those found on GBIF; however, they differ from the GBIF data in that the collection, identification and pathogen data are presented in separate tables and can be linked via sampleIDs and subsampleIDs. Tick absences are represented in NEON data when a given collection record contains no sampleID or corresponding identification record, whereas this is not the case in the GBIF dataset. Column names also differ between GBIF and NEON data, including the use of an individualCount column in NEON data that is absent in GBIF data. The NEON tick identification data are thus presented in a wider format, with one entry per tick species, sex, and life stage in a given collection event, whereas the GBIF data are in a longer format, with one record for each individual tick.

Numerous other related datasets, including those on organisms that may serve as tick hosts are also available from the NEON data portal [20]. These datasets describe tick-borne pathogens in small mammals [34] as well as small mammal diversity and abundance [35] and breeding land bird diversity and abundance [36]. These data are collected at the same NEON sites where tick data collection occurs, with the specific plots and exact dates of sampling being distinct across data products. Small mammal abundance and diversity are collected using box trap sampling with 4–6 bouts of sampling per site spaced 1–2 months apart. One bout consists of trapping 3–8 grids (10 m × 10 m) per site, and three of those grids are sampled for three nights, while the remainder are sampled for one night [37]. Since 2020, ear and blood samples collected from these rodents have been tested for tick-borne pathogens, while prior to 2020 rodent blood samples were tested for hantavirus antibodies. Birds are sampled using 6-minute point counts at 5–25 grids within each site, and data on the species, distance, and sex are recorded. Bird sampling occurs once or twice per season at small and large sites, respectively [38]. Several R packages (such as neonUtilities [39]) and free tutorials have been developed by NEON to facilitate working with these interoperable and open-access datasets.

## EDITOR'S NOTE

This paper is part of a series of Data Release articles working with GBIF and supported by the Special Programme for Research and Training in Tropical Diseases (TDR), hosted at the World Health Organization [40].

## DECLARATIONS
## LIST OF ABBREVIATIONS

CDC: Centers for Disease Control and Prevention; GBIF: Global Biodiversity Information Facility; GUAN: Guanica Forest site; LiDAR: light detection and ranging; NEON: National Ecological Observatory Network; NDVI: Normalized Difference Vegetation Index; NLCD: National Landcover Database; PCR: polymerase chain reaction; PDE: percent difference in enumeration; PTD: percent taxonomic difference; TDR: Special Programme for Research and Training in Tropical Diseases.

## ETHICS APPROVAL

Not applicable.

## CONSENT FOR PUBLICATION

Not applicable.

## COMPETING INTERESTS

The authors declare that they have no competing interests.

## FUNDING

NEON is a project sponsored by the National Science Foundation. The work described here was funded by National Science Foundation grant NSF 1724433.

## AUTHORS' CONTRIBUTIONS

KT was responsible for project administration; KT and SP contributed to methodology and data curation; SP, KT and AB prepared the dataset for GBIF; SP wrote the manuscript. All authors read and approved the final version of the manuscript.

## ACKNOWLEDGEMENTS

The National Ecological Observatory Network is a program sponsored by the National Science Foundation and operated under cooperative agreement by Battelle. This material is based in part upon work supported by the National Science Foundation through the NEON Program.

The design and implementation of the NEON tick collection and pathogen testing protocols have benefitted from the input of several scientists over the years. We would like to thank the numerous members of the NEON Technical Working Groups for ticks and tick-borne parasites for their insights into protocol design and data collection. We would also like to acknowledge the field scientists and technicians who implemented the protocols in varied locations across the USA. Past and current NEON scientists have also contributed to the project design and data quality. Finally, Lorenza Beati with the US National Tick Collection has overseen all tick identifications, and Stephen Rich with the Laboratory of Medical Zoology has overseen tick pathogen testing for the data included here. Any use of trade, firm, or product names is for descriptive purposes only and does not imply endorsement by the US Government.

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
