## [Reviewer Report]

Reviewer name and names of any other individual's who aided in reviewer Benjamin CullDo you understand and agree to our policy of having open and named reviews, and having your review included with the published papers. (If no, please inform the editor that you cannot review this manuscript.)YesIs the language of sufficient quality?YesPlease add additional comments on language quality to clarify if needed
Are all data available and do they match the descriptions in the paper? YesAdditional CommentsAre the data and metadata consistent with relevant minimum information or reporting standards? See GigaDB checklists for examples <a href="http://gigadb.org/site/guide" target="_blank">http://gigadb.org/site/guide</a>YesAdditional CommentsIs the data acquisition clear, complete and methodologically sound?YesAdditional CommentsIs there sufficient detail in the methods and data-processing steps to allow reproduction?YesAdditional Commentsline 167-169 - this volumetric method for estimating large numbers of larvae is very interesting and would be valuable to other tick researchers I think. Please could you provide a reference (if available) or slightly more detail on how it the method performed. Is there sufficient data validation and statistical analyses of data quality? YesAdditional CommentsIs the validation suitable for this type of data?YesAdditional CommentsIs there sufficient information for others to reuse this dataset or integrate it with other data?YesAdditional CommentsAny Additional Overall Comments to the AuthorThis is a valuable long-term dataset on ticks that has potential to be combined with other available datasets for various analyses into multiple factors related to ticks and tick-borne diseases. This will be of use to researchers and scientists across multiple disciplines. I only have a minor recommendation to make, and noticed a few minor spelling errors: line 25 - should say " ...relative importance of other potential drivers ..." line 155 - repellent line 183 Rickettsia rickettsiiRecommendationMinor Revision

---

## [Reviewer Report]

Reviewer name and names of any other individual's who aided in reviewer Marlon E. CobosDo you understand and agree to our policy of having open and named reviews, and having your review included with the published papers. (If no, please inform the editor that you cannot review this manuscript.)YesIs the language of sufficient quality?YesPlease add additional comments on language quality to clarify if needed
Are all data available and do they match the descriptions in the paper? YesAdditional CommentsAre the data and metadata consistent with relevant minimum information or reporting standards? See GigaDB checklists for examples <a href="http://gigadb.org/site/guide" target="_blank">http://gigadb.org/site/guide</a>YesAdditional CommentsIs the data acquisition clear, complete and methodologically sound?YesAdditional CommentsIs there sufficient detail in the methods and data-processing steps to allow reproduction?NoAdditional CommentsNo required for this type of publication or the goal of this contribution.Is there sufficient data validation and statistical analyses of data quality? NoAdditional CommentsNo required for this type of publication or the goal of this contribution.Is the validation suitable for this type of data?YesAdditional CommentsIs there sufficient information for others to reuse this dataset or integrate it with other data?YesAdditional CommentsAny Additional Overall Comments to the AuthorDear authors,  The manuscript entitled "Tick abundance, diversity and pathogen data collected by the National Ecological Observatory Network" presents interesting information o the tick data from NEON. I consider that this is a fine contribution that could help researchers interested in the topic to understand relevant details on the origin and quality of the data presented.   Please see below my minor comments.  Minor comments: 1. Please explain the reason why only nymphs were considered in tests for pathogen detection. 2. Recently I used the neonUtilities R package. I found it really useful to get and organize data from NEON. I think the authors could mention this tool in the manuscript (perhaps somewhere close to where other datasets are mentioned). 3. Please add a reference to having a clear idea of where the eco-climatic divisions were obtained from.RecommendationMinor Revision

---

## [Reviewer Report]

Upload additional filesDRR-202202-08/form/GigaByte_Review_DRR_202202_08_Tick_CJA_includes_Data_Review_08Mar2022.pdfReviewer name and names of any other individual's who aided in reviewer Chris ArmitDo you understand and agree to our policy of having open and named reviews, and having your review included with the published papers. (If no, please inform the editor that you cannot review this manuscript.)YesIs the language of sufficient quality?YesPlease add additional comments on language quality to clarify if needed
Are all data available and do they match the descriptions in the paper? NoAdditional Comments1. There is reference to NEON datasets in the Data Availability section of the manuscript. However, the relationship between the GBIF dataset and the various NEON datasets is very confusing. For example, consider the following statements in the Data Availability section of the manuscript.  • “These data along with other related datasets are also available in a different format from the NEON data portal.”  • Closely related NEON datasets describing tick-borne pathogens in small mammals (https://doi.org/10.48443/6pfn-t955) as well as small mammal diversity and abundance (https://doi.org/10.48443/h3dk-3a71) and breeding landbird diversity and abundance (https://doi.org/10.48443/88sy-ah40) that may serve as tick hosts are also available for the same sites and an overlapping time-frame.  2. There are no protozoan parasites listed in the GBIF SIMPLE table.  I recommend that the authors update the Data Availability section of the manuscript so that it unambiguously describes the Dataset that is detailed in the manuscript. In addition, I request that the authors confirm whether protozoan parasites were identified in this study, and that they update the GBIF record accordingly.Are the data and metadata consistent with relevant minimum information or reporting standards? See GigaDB checklists for examples <a href="http://gigadb.org/site/guide" target="_blank">http://gigadb.org/site/guide</a>YesAdditional CommentsIs the data acquisition clear, complete and methodologically sound?YesAdditional CommentsIs there sufficient detail in the methods and data-processing steps to allow reproduction?YesAdditional CommentsIs there sufficient data validation and statistical analyses of data quality? YesAdditional CommentsIs the validation suitable for this type of data?YesAdditional CommentsIs there sufficient information for others to reuse this dataset or integrate it with other data?YesAdditional CommentsAny Additional Overall Comments to the AuthorThis Data Release manuscript describes tick abundance in the USA. The manuscript is well written, the dataset is in English, and the Metadata consists of 428,960 human observations of sampled ticks.  The GBIF summary table with the Metadata was included in the Darwin Core DwC package. Of note, the 2,757 Latitude / Longitude Geographic Locations are listed in the GBIF Darwin Core DwC package “event.txt” file rather than the GBIF Darwin Core DwC package “occurrence.txt” file. However, GBIF have been particularly helpful by integrating the contents of these two files into a GBIF SIMPLE table that can be downloaded from GBIF and that includes all 428,960 human observations plus accompanying Latitude / Longitude Geographic Locations, Country Code, Event Dates, and Taxonomic Keys   Of note, in the GBIF Dataset Description, it states the following:  • “A subset of identified nymphal ticks are tested for the presence of bacterial and protozoan pathogens”  However, the SIMPLE table that can be downloaded from GBIF only includes Ixodida (ticks) and bacterial pathogens of ticks. Consequently, protozoan pathogens are missing from the GBIF dataset. Of the 428,960 human observations of sampled ticks, there are a total of 92,645 samples that have been identified at the species level.  The License for this Dataset is: CC-0 1.0  The major issues with this Dataset are as follows:  1. There is reference to NEON datasets in the Data Availability section of the manuscript. However, the relationship between the GBIF dataset and the various NEON datasets is very confusing. For example, consider the following statements in the Data Availability section of the manuscript.  • “These data along with other related datasets are also available in a different format from the NEON data portal.”  • Closely related NEON datasets describing tick-borne pathogens in small mammals (https://doi.org/10.48443/6pfn-t955) as well as small mammal diversity and abundance (https://doi.org/10.48443/h3dk-3a71) and breeding landbird diversity and abundance (https://doi.org/10.48443/88sy-ah40) that may serve as tick hosts are also available for the same sites and an overlapping time-frame.  2. There are no protozoan parasites listed in the GBIF SIMPLE table.  I recommend that the authors update the Data Availability section of the manuscript so that it unambiguously describes the Dataset that is detailed in the manuscript. In addition, I request that the authors confirm whether protozoan parasites were identified in this study, and that they update the GBIF record accordingly.
RecommendationMinor Revision